# *Select, Prompt, Filter*: Distilling Large Language Models for Summarizing Conversations

**Minh-Quang Pham**,* **Sathish Reddy Indurthi**,*
**Shamil Chollampatt, Marco Turchi**
Zoom Video Communications
minhquang.pham@zoom.us, sathishreddy.indurthi@zoom.us,
shamil.chollampatt@zoom.us, marco.turchi@zoom.us

## Abstract

Large language models (LLMs) like ChatGPT can be expensive to train, deploy, and use for specific natural language generation tasks such as text summarization and for certain domains. A promising alternative is to fine-tune relatively smaller language models (LMs) on a particular task using high-quality, in-domain datasets. However, it can be prohibitively expensive to get such high-quality training data. This issue has been mitigated by generating weakly supervised data via *knowledge distillation* (KD) of LLMs. We propose a three-step approach to distill ChatGPT and fine-tune smaller LMs for summarizing forum conversations. More specifically, we design a method to selectively sample a large unannotated corpus of forum conversation using a semantic similarity metric. Then, we use the same metric to retrieve suitable prompts for ChatGPT from a small annotated validation set in the same domain. The generated dataset is then filtered to remove low-quality instances. Our proposed *select-prompt-filter* KD approach leads to significant improvements of up to 6.6 ROUGE-2 score by leveraging sufficient in-domain pseudo-labelled data, over a standard KD approach given the same size of training data.

## 1 Introduction

Large language models (LLMs) such as the GPT-series models have demonstrated great strengths in a range of natural language understanding and generation tasks with their ability to do *few-shot* or *in-context learning* (ICL, Brown et al., 2020). In this scenario, the input for a desired task is enhanced by incorporating a few examples (or *demonstrations*) of that particular task given as the *prompt* to the LLM. This helps to customize and optimize the LLM at inference time, leading to higher-quality results. Many of these LLMs, however, are astronomically expensive and environmentally unsustainable to train and use (Bender et al., 2021; Wu

---
*Equal contribution

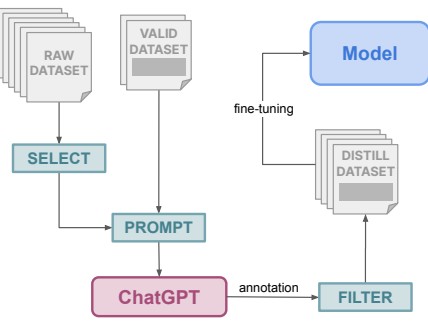

Figure 1: Schematic diagram of select-prompt-filter knowledge distillation.

et al., 2022; Li et al., 2023). An alternate strategy is to fine-tune relatively much smaller language models (LM) like BART (Lewis et al., 2020) or T5 (Raffel et al., 2020) on the target task. These fine-tuned models, however, require large annotated training datasets to achieve competitive results, preventing the adoption of this approach for a wide variety of tasks and domains. Obtaining sufficient annotated data from human annotators can also be substantially expensive (Wang et al., 2021).

An approach to address this issue of limited high-quality annotated data involves creating synthetic data that extracts and distills knowledge from ICL-capable LLMs to smaller specialized models. This technique of knowledge distillation (Hinton et al., 2015; Kim and Rush, 2016) helps specialize smaller pre-trained LMs to particular tasks using large amounts of such synthetic annotations. For example, Wang et al. (2021) use GPT-3 to annotate training data for nine NLP tasks, including summarization, estimating up to 96% reduction in costs over using crowdsourcing and achieving similar performance. Ding et al. (2022) use GPT-3 to get labels and even generate data from scratch for sequence tagging tasks. However, previous work on distillation addresses the data annotation process without careful data selection, prompt engineering,

or data filtering, leading to suboptimal results.

To get a better-distilled model and optimize the cost of synthetic data annotation, we propose a systematic distillation approach, named *select-prompt-filter knowledge distillation* (SPF KD), for summarization. We use ChatGPT as the ICL-capable LLM that we distill. In this paper, we focus on summarizing online messaging conversations (e.g., forums) for which human-annotated datasets are not publicly available, though our method can be applied to other low-resource domains as well. First, a suitable subset of conversations is *selected* for distillation from a large unannotated raw corpus, based on similarity to a small annotated validation set. Then, for each conversation in the selected subset, semantically similar conversations are retrieved from the validation set and sent to ChatGPT as the prompt to get its synthetic summary. Liu et al. (2022a) use a similar prompt-retrieval approach based on various sentence embeddings to improve GPT-3 performance on downstream tasks, although not in the context of synthetic data annotation. Thereafter, we apply a *filter* on the generated conversation-summary pairs to remove spurious and low-quality annotations. Figure 1 shows a schematic overview of our approach. Our proposed method uses a semantic similarity score (Cer et al., 2018) and a reference-less evaluation measure (Egan et al., 2022) at different steps. We evaluate our approach on three forum and email conversation test sets. We show that the proposed KD method is able to significantly improve summarization performance of up to 6.6 ROUGE-2 scores compared to a standard KD approach.

## 2 Approach

Our method addresses the lack of high-quality annotated summarization datasets on certain domains to fine-tune smaller summarization models via sequence-level knowledge distillation (Kim and Rush, 2016). We propose a three-step approach to get the annotated training data from ICL-capable LLMs like ChatGPT, which acts as our *teacher* model that we distill. The three steps (Figure 2) in our proposed *select-prompt-filter* (SPF) knowledge distillation approach are (i) **data selection** (ii) **prompt retrieval**, and (iii) **data filtering**. These steps are done to get high-quality synthetic summaries for the target task of summarizing conversations. The synthetic dataset will then be used to fine-tune a smaller pretrained LM (*student* model).

Fundamental to our approach are two metrics that we use in the above steps:

**USE-Cosine**: We use Universal Sentence Encoder[1] (USE, Cer et al. 2018) to get the embeddings of a conversation given as a text string. USE internally uses a Deep Averaging Network (Iyyer et al., 2015) that averages the embeddings of the words and bigrams and passes it through a feed-forward network. We compute the cosine distance between the two conversation embeddings. We refer to this metric as *USE-cosine*, which we use for computing *conversation-conversation* similarity in the selection and prompt retrieval steps.

**Shannon Score**: Shannon Score[2] (Egan et al., 2022) is a reference-free summary evaluation metric that measures the difference in the ability of a language model to generate the conversation $\mathcal{C}$ with and without the summary $\mathcal{S}$. Specifically, GPT-2 is used to compute the *information difference*, i.e., the difference between log probabilities of generating $\mathcal{C}$ given $\mathcal{S}$ as the prompt and generating $\mathcal{C}$ without it. The final metric, Shannon Score ($SS$), is the ratio of this value to the information difference when the $C$ itself is used as the summary. This is under the assumption that the conversation conveys the entire information and thus the denominator becomes the upper bound of the information difference.

$$SS(\mathcal{C}, \mathcal{S}) = \frac{\log P(\mathcal{C}|\mathcal{S}) - \log P(\mathcal{C})}{\log P(\mathcal{C}|\mathcal{C}) - \log P(\mathcal{C})} \quad (1)$$

We use Shannon Score to compute the *conversation-summary* relevance and filter the synthetic annotations generated by ChatGPT that do not satisfy a certain threshold.

In our proposed approach, we rely on the availability of a large unannotated training dataset of forum threads ($D_t$) for knowledge distillation for our target task. We use a large corpus of forum conversations for this. We also assume the availability of a small annotated validation set $D_v$ with $M$ conversation-summary pairs. We describe the three steps used in our approach in greater detail below.

### 2.1 Data Selection

To control costs when using ChatGPT for annotation, we use a similarity-based selection approach to select $N$ conversations from $D_t$ for generating synthetic summaries. As shown in Figure 2 (left),

[1] https://tfhub.dev/google/universal-sentence-encoder/4
[2] https://github.com/PrimerAI/blanc/

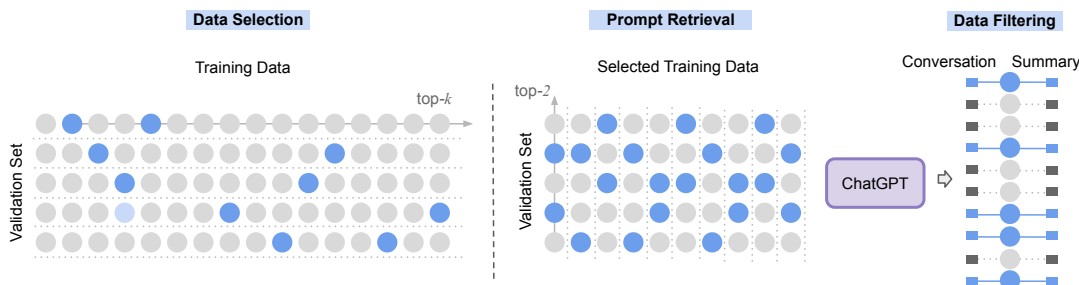

Figure 2: The three steps of our proposed approach: (i) data selection, (ii) prompt retrieval, and (iii) data filtering.

for each validation conversation in $D_v$ we select top $k$ conversations ($k \approx N/M$) from $D_t$ without replacement using the USE-cosine similarity. In this way, we ensure that we represent every example of the validation set independently to capture maximum variance in the training data. $N$ is determined based on the available budget. This selected training data is denoted by $D_s$, which is a subset of $D_t$.

## 2.2 Prompt Retrieval

For each conversation in the selected training data $D_s$, we apply a similarity-based prompt retrieval technique similar to Liu et al. (2022c) to construct the prompt. We select one or more examples from the $D_v$ that are closest to each conversation in $D_s$ according to the USE-cosine metric (see Figure 2, center). The retrieved examples along with the corresponding training conversation to be summarized are passed to ChatGPT as the prompt to get the summary (see example prompt in Appendix A.1). Thus, we get an annotated training set $D_s^*$ which is a set of forum threads and their corresponding ChatGPT generated summaries.

## 2.3 Data Filtering

Some of the ChatGPT-generated summaries in $D_s^*$ can be of poor quality and diverge from the conversation significantly. We filter out such spurious examples that do not satisfy the certain threshold of Shannon Score, to get a smaller annotated training dataset $D_f^*$ (see Figure 2, right). $D_f^*$ is the distilled training data from the *teacher* that is then used to finetune a *student* model specialized for summarizing conversations.

## 3 Experimental Setup

### 3.1 Data

We use conversations from StackExchange forums[3], except non-English language forums and StackOverflow due to its size, as the raw unannotated dataset. For validation and testing, we use the Ubuntu and NYC forum summarization corpora (Bhatia et al. 2014, 100 instances each), and the BC3 email summarization corpus (Ulrich et al. 2008, 40 instances). We divide each dataset equally for validation and testing. We combine the three validation sets into a single validation set of 120 instances.

We use OpenAI ChatGPT as our teacher model for distillation. We restrict to selecting 100K conversations for ChatGPT annotation from StackExchange to demonstrate a limited budget scenario. Annotating 100K conversations costs about 200 USD[4] (gpt-3.5-turbo) assuming each annotation input is under 1000 tokens. Our baseline is a standard knowledge distillation (Standard KD) approach, where a random 100K subset is selected from StackExchange for distillation. For the baseline, we select two random examples as the prompt from our validation set to be sent to ChatGPT for getting synthetic summaries and we do not filter the results. During the data selection step of our proposed SPF KD approach, for each conversation in our validation set, we select $k = 833$ examples ($\frac{100000}{120} \approx 833$) from StackExchange based on USE-cosine similarity (Section 2.1). For each selected conversation, we prompt ChatGPT with two examples from the validation set that are closest to it based on the USE-Cosine measure. For data filtering, we set a Shannon Score threshold of 0.15 to retain at least 50% of annotated data.

---

[3]https://archive.org/download/stackexchange
[4]https://openai.com/pricing

|  | Ubuntu | | | NYC | | | BC3 | | |
|---|---|---|---|---|---|---|---|---|---|
|  | R-1 | R-2 | R-L | R-1 | R-2 | R-L | R-1 | R-2 | R-L |
| ChatGPT (zero-shot) | 39.9 | 17.4 | 29.1 | 40.7 | 14.5 | 26.2 | 43.6 | 14.5 | 27.0 |
| ChatGPT (one-shot, random) | 44.6 | 21.2 | 31.9 | **43.3** | 16.0 | 27.5 | 43.3 | 15.2 | 28.3 |
| ChatGPT (two-shot, random) | 47.3 | 24.4 | 36.0 | 42.5 | 16.4 | 27.9 | 44.8 | 16.2 | 28.5 |
| ChatGPT (two-shot, USE-cosine) | 49.0 | 25.9 | 37.2 | 42.9 | 16.6 | 28.3 | 46.9 | 16.0 | 29.9 |
| PEGASUS (Zhang et al., 2020) | 27.9 | 17.0 | 23.0 | 32.0 | 10.6 | 19.9 | 27.8 | 7.8 | 16.6 |
| BRIO (Liu et al., 2022b) | 33.4 | 17.6 | 24.0 | 35.0 | 12.0 | 21.0 | 32.3 | 10.8 | 20.0 |
| Standard KD | 48.8 | 26.8 | 37.5 | 41.5 | 17.3 | 27.7 | 44.7 | 17.3 | 29.9 |
| SPF KD | **55.4** | **33.4** | **43.5** | 41.7 | **18.2** | **28.3** | **47.2** | **19.8** | **32.3** |

Table 1: Performance of our proposed select-prompt-filter (SPF) KD and baselines.

## 3.2 Model

For both Standard and SPF KD, we fine-tune one student model each with the corresponding distillation data and evaluate on all three test sets. Similar to the findings of Demeter et al. (2023), our preliminary investigations indicated that a fine-tuned BART model produces better-quality summaries compared to fine-tuning models such as PEGASUS (Zhang et al., 2020) Hence, we use an XSum-finetuned (Narayan et al., 2018) BART-large[5] (406.3M params, Lewis et al. 2020) as the student model for both KD approaches. For each of our KD experiments, we further fine-tune this BART model on the corresponding distilled dataset for 5 epochs, beyond which we observed overfitting on our validation set. We used a batch size 16 and learning rate 2e-5.

We report results using various prompting strategies for ChatGPT: with no priming examples (*zero-shot*), with one or two randomly-selected priming example(s) from the validation set (*one or two-shot, random*), and with two validation examples closest to the test conversation based on USE-cosine measure (*two-shot, USE-cosine*). ChatGPT (two-shot, random) is the *teacher* model for Standard KD due to the randomly selected prompting examples, and ChatGPT (two-shot, USE-cosine) is the *teacher* model in SPF KD as it incorporates the prompt retrieval step. We also include results of two competitive off-the-shelf summarization models: PE-GASUS[6] (570.8M params, Zhang et al. 2020) and BRIO[7] (406.3M params, Liu et al. 2022b).

[5] https://hf.co/facebook/bart-large-xsum
[6] https://hf.co/google/pegasus-large
[7] https://hf.co/Yale-LILY/brio-cnndm-uncased

## 4 Results and Analysis

In Table 1, we report the results of the proposed SPF-KD approach compared to standard KD, both using a pre-trained BART model fine-tuned on annotations generated by ChatGPT. Our baseline, standard KD, outperforms PEGASUS (Zhang et al., 2020), and BRIO (Liu et al., 2022b) showing the efficacy of KD and the importance of adding Stack-Exchange data. Compared to standard KD, our SPF KD achieves substantial gains of 6.6, 0.9, and 2.5 ROUGE-2 scores on Ubuntu, NYC, and BC3, respectively, showing the strength of our proposed approach. The larger magnitude of improvement on Ubuntu can be attributed to a large proportion of similar forums in our StackExchange dataset that our data selection step may have over-represented. Similar to Wang et al. (2021), we also observe that KD using ChatGPT summaries outperforms the corresponding ChatGPT teacher model itself. We attribute this to fine-tuning the smaller model on conversational (forums) data and being specialized to do only the summarization task. We provide a few examples of the outputs of the models in Appendix A.2. This makes KD, especially our SPF KD, an effective and cheap strategy to build specialized summarization models without relying on external services.

### 4.1 Ablation study

We tease apart the contribution of each step of our proposed approach to the overall performance improvement over the standard KD approach (see Table 2). We find that each of the three steps improves on the three datasets. Moderate gains are observed in NYC on adding each step. The data selection step notably improves BC3. Data filtering consistently improves performance across the three test sets. The biggest improvements are

|                    | Ubuntu | NYC  | BC3  |
|--------------------|--------|------|------|
| Standard KD        | 26.8   | 17.3 | 17.3 |
| + Data Selection   | 28.5   | 17.4 | 19.7 |
| + Prompt Retrieval | 32.4   | 17.8 | 19.7 |
| + Data Filtering   | **33.4** | **18.2** | **19.8** |

Table 2: ROUGE-2 results of using each step in the select-prompt-filter KD approach.

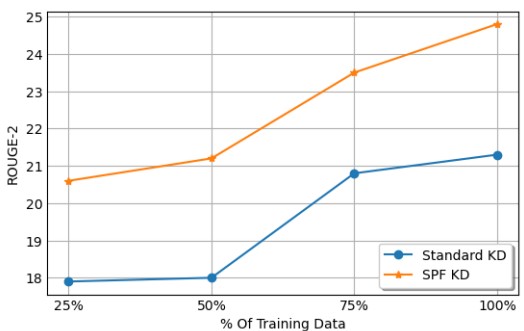

Figure 3: Average ROUGE-2 scores vs. percentage of distillation data used

observed in the Ubuntu test set at each step. We attribute this to the abundance of Ubuntu-related forums in StackExchange which are adequately exploited in our approach.

### 4.2 Distillation Cost

We study the performance of the standard KD (Baseline) and our SPF KD approach under lower ChatGPT-annotation budgets. To do this, we train the student model using randomly sampled subsets of varying sizes of the distillation data obtained using both Standard KD and SPF KD (after filtering) and compute the average ROUGE-2 across the three test sets. This is plotted in Figure 3. We find that fine-tuning the student model with 50% of SPF KD training data achieves a similar ROUGE-2 score compared to utilizing the entire Standard KD training data.

### 4.3 Domain Relevance

We can use the aggregated USE-cosine metrics to estimate the relevance of the selected training data for each domain represented by each validation set. For every validation example in the three domains (Ubuntu, NYC, and BC3), we compute the mean USE-cosine across the 833 corresponding selected training examples. We then average this value across all validation examples in the corresponding domain. We find that we get 0.7 for

Ubuntu, 0.4 for NYC, and 0.5 for BC3. This shows we get more similar training examples for Ubuntu compared to the other two domains. Interestingly, the final ROUGE-2 gains using our SPF-KD approach also follow the same ranking as this aggregated USE-cosine measure (Ubuntu > BC3 > NYC).

## 5 Conclusion

We propose a methodical knowledge distillation approach that finetunes a smaller LM for summarizing conversations with synthetic summaries generated by ChatGPT. Our approach selects a suitable subset to be distilled, retrieves matching prompts for ChatGPT, and filters the ChatGPT-generated summaries to yield better-quality training data. Our proposed method substantially improves over a standard distillation approach on three test sets covering forum and email conversations.

## Limitations

The limitations of our paper are:

- Our test sets are relatively small, with only 120 samples. However, this limitation is due to the lack of data in the conversational scenario.

- Our experiments have been run using 100K training points while having access to 7 million unlabelled data points.

- We do not explore a wide variety of prompting strategies.

- We only validate our data-centric approach on the BART-large model while it can be applied to any large pretrained language model.

- We only use metrics from the ROUGE family to evaluate models in our experiments.

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

# A  Appendix

## A.1  Prompt Structure

The format of the prompt given to ChatGPT is given below:

```
Given one example of summarization:
Conversation:
<example_conversation>
Summary:
<example_summary>
***
Please summarize the following conversation:
<conversation>
```

We pass a prompt in the above format as part of the user's input to ChatGPT to get a response from it.

### A.2 System Outputs

In Table 3 and Table 4 we show output summaries generated by ChatGPT (two-shot, USE-cosine), Standard KD, and SPF KD.

| Conversation | Summary |
|---|---|
| **<userA>** : I'm driving from Cleveland. I want to stay downtown and see a few sights on Saturday, drive to the game on Sunday (unless there's a shuttle?), then head to the casino or Niagara Falls. How far is Seneca from downtown Buffalo? Are there any other casinos? How far is Niagara Falls? Can I see enough from the US side, or should I suck it up and get a passport and go into Canada? Any recommendations would be appreciated.
**<userB>** : You can pick up a special Game Day bus to Ralph Wilson Stadium: http://www.nfta.com/metro/gameday-bills.asp Niagara Falls is about 20 minutes from downtown Buffalo. You can enjoy the falls from the U.S. side, but if you want to go see the Canadian falls, you won't need a passport – all you'll need is a birth certificate and a government-issued photo ID. I'll let someone else answer your casino questions.
**<userA>** : Thanks. Can you recommend a hotel that is close to the metro station downtown so we can hop the metro to go to the game? The game is November 17—-what weather should I expect?
**<userD>** : You can expect the same weather in Buffalo as you get in Cleveland. We get your weather probably the day after you get it (from the west). It could rain, it could snow, it could be a nice, sunny day... who knows Regarding hotels downtown, here's a list from a previous post: 1) Hampton Inn tripadvisor.comHotel_Review-g60974-d224389-... 2) Adam's Mark tripadvisor.comHotel_Review-g60974-d93115-R... 3) Hyatt Regency Hotel tripadvisor.comHotel_Review-g60974-d93122-R... 4) Comfort Suites tripadvisor.comHotel_Review-g60974-d93128-R... You can catch the train on Main Street and it's free as long as you stay above ground. The bus station where you will most likely catch the shuttle to the game is near Main and N. Division St, across from Church St. Niagara Falls: The Seneca Niagara Casino is within walking distance of the actual Falls. If it's a nice day and you feel like walking, it's an easy walk to the Canadian side across the Rainbow Bridge. Just bring you birth cert and driver's license so they let you back into the USA. There is a casino in downtown Bflo, but it's nothing special, from what I've been told.
**<userE>** : the metro train does not go out to ralph wilson stadium. the stadium is in orchard park, about 25 min south of buffalo. (you'll pass it on the 90 as you drive up)if you drive to the game, leave your hotel early as traffic can get thick. consider driving, tailgating is art in buffalo! it's a lot of fun to experience. i would recommend the hampton inn. good central location for night life.
**<userF>** : Don't waste your time or money at the Canadian casinos. The slots are VERY tight and you will never see a minimum below 15/20$. Also, if you're a smoker, they are non-smoking and do NOT offer free drinks, whereas the Seneca casino on the US side does. Go Bills! Boo Browns!
**<userG>** : No offense to Gear but the Canadian Casinos are far superior to the Seneca if your motivation is a nice atmosphere, shopping, and beauty. I can not speak for the gambling outcomes. To the Seneca's credit , they have great restaurants and a very nice concert venue. If you are into casinos, I would check out both....
**<userA>** : Thanks for all of your advice. We went ahead and booked the Country Inn near the Falls in Canada for 3 nights (I figure we can check out the Falls since the only time I was there was when I was a toddler). We'll drive back on Monday and check in to the hotel in Hamburg (Holiday Inn....cheap) and then go over to the stadium early for some tailgating. It looks like it's only about 5 miles away, so even with crazy traffic I'm hoping its not too big of a headache. Is alcohol permitted in the parking lot? Little gas grills? And what about the restroom situation—-are there port-o-johns available?? And how soon do they open the stadium? Thanks so much for all of your help!
**<userG>** : drinks-yes, grills-yes, the lots open at the crack of dawn....it is full contact tailgating in Buffalo and I think after experiencing it, you will change your team affiliation. After going to a few Browns games...I believe Buffalo has the tailgating title...but it is very close.....Go Toronto...oh I mean Buffalo....!!" | **ChatGPT:** The conversation is about a person who is planning to visit Buffalo to attend a Sabres game. They were looking for recommendations for restaurants near their hotel. Participants made several recommendations, including Salsarita, Papaya, Buffalo Chop House, and E.B. Green's. They also asked for advice on the free train that could take them to the arena and back to their hotel. They were also interested in visiting the casino or Niagara Falls. There was a discussion about the Canadian Casinos versus the Seneca casino on the US side. They also wanted to know about tailgating and if alcohol is permitted in the parking lot.

**Standard KD:** The conversation is about a person who wants to go to Buffalo for a Buffalo Bills game and wants to know how far it is from downtown Buffalo, if there are any other casinos, how far is Niagara Falls, and if they can see enough from the US side or if they should get a passport and go into Canada. *<userB>* recommends a special Game Day bus to Ralph Wilson Stadium and Niagara Falls. *<userD>* recommends hotels downtown, including Hampton Inn, Adam's Mark, Hyatt Regency Hotel, and Comfort Suites. The Seneca Niagara Casino is within walking distance of the actual Falls and there is a casino in downtown Buffalo. *<userF>* advises against going to Canadian casinos as they are tight and do not offer free drinks. *<userG>* recommends checking out both Canadian and Seneca casinos. *<userA>* thanks everyone for their advice and decides to book the Country Inn near the Falls in Canada for three nights. They will drive back on Monday and check in to the hotel in Hamburg and then go over to the stadium early for some tailgating.

**SPF KD:** A person is driving from Cleveland to Buffalo and wants to see a few sights on Saturday, drive to the game on Sunday, then head to the casino or Niagara Falls. The person wants to know how far Seneca is from downtown Buffalo, if there are any other casinos, and if they can see enough from the US side, or should they get a passport and go into Canada. A reply suggests that the person can pick up a special Game Day bus to Ralph Wilson Stadium, and Niagara Falls is about 20 minutes from downtown Buffalo. The reply also recommends a hotel that is close to the metro station downtown, and the weather in Buffalo is the same as in Cleveland. The Seneca Niagara Casino is within walking distance of the actual Falls. If it's a nice day and the person feels like walking, it's an easy walk to the Canadian side across the Rainbow Bridge. There is a casino in downtown Buffalo but it's nothing special. Another reply suggests leaving the hotel early as traffic can get thick. They recommend the Hampton Inn, which is a good central location for night life. The Canadian casinos are far superior to the Seneca if the person's motivation is a nice atmoshphere, shopping, and beauty.

**Reference:** A person was driving from Cleveland to Buffalo to attend the game on Sunday. Further he was planning to go casino or Niagara Falls. He wanted to how far be Seneca from downtown Buffalo, were there any other casinos, how far was Niagara Falls and was passport required to go into Canada? Someone replied that he can pick up a special Game Day bus to Ralph Wilson Stadium. Niagara Falls was about 20 minutes from downtown Buffalo. To travel to Canadian side Niagara fall, birth certificate and a government-issued photo ID will be needed. The other reply was that there were many hotels in the downtown. The trains and buses were available for transportation. The Seneca Niagara Casino was within walking distance of the actual fall. The other suggestion was not to waste time or money at the Canadian casinos. There were some disagreements over the quality of the Canadian casinos. |

Table 3: Summaries generated by ChatGPT (two-shot, USE-cosine), Standard KD, and SPF KD on an example NYC forum conversation (usernames are anonymized).

| Conversation | Summary |
|---|---|
| **<userA>**: Since yesterday (when I installed the latest Ubuntu 'edgy' kernel security update), my Serval Performance system has failed to boot several times. This problem required a hard shutdown and reboot. The first time this occurred, I tried booting into recovery mode, which seemed to work around the issue. The second time, even recovery mode did not work; three times I attempting to boot without success. Since I could see the boot status this time I noticed the message ""BUG: soft lockup on CPU#0"". This did not sound very good to me. Fortunately, I have just booted my system successfully again after flipping the wireless card switch on the chassis to the 'on' position. I had previously turned it 'off' since I was not using it at the time and had noticed an apparent CPU bug whereby one of my cores was always at 100% frequency even when not in use according to the system monitor. Anyhow, I don't know if the switch was the cause/cure or not, because when the problem first began I still managed to boot with the switch 'off' and so far I have only had one success with it 'on'. (I need to do some important work before I can test this theory and risk not being able to boot again.) Has anyone else heard of or experienced a similar problem? -<userA>- 

 **<userB>**: Hi <userA> - I have one other report of ""BUG: soft lockup on CPU#0"" although it was a different machine. I'm trying to re-create the problem on a Serval now and will get back to you shortly. 
 **<userC>**: Perhaps it was caused by this? http://ubuntuforums.org/showthread.php?t=318206 
 **<userB>**: <userA>- we're not experiencing the problem using nVidia's driver from the repos - does hackmasters thread look relevant? 

 **<userA>**: I do have the driver version mentioned in that notice. However, neither of the files noted are missing from /usr/lib/xorg/modules. Also, my boot process was failing well before attempting to initialize X. I now reinstalled the nvidia-glx package just in case. Since flipping the wireless card switch back to the 'on' position, I have not had this problem. I haven't yet tried switching it 'off' again to see if I can reproduce it. Regarding the issue of one cpu core running at 100% frequency all the time, my best guess now is that it may not be related to the wireless card/driver. I have recently noticed that it the issue seems to be related to coming out of hibernation. If I boot my system fresh, I do not observe the behavior, but if I hibernate and resume then I notice it. Many thanks to Carl and hackmeister for your help. If you happen to discover any more information about the error message I initially posted, let me know. Otherwise, I have my workaround for now. -<userA>- | **ChatGPT:** The user's Serval Performance system failed to boot several times after installing the latest Ubuntu 'edgy' kernel security update. The user noticed the message "BUG: soft lockup on CPU#0". The user tried booting into recovery mode, which worked the first time but not the second time. The user turned on the wireless card switch, and the problem stopped occurring. The user also noticed that one CPU core was always at 100% frequency when coming out of hibernation. The user reinstalled the nvidia-glx package. The problem seemed to be related to hibernation. Another user suggested that the problem could be caused by missing files in /usr/lib/xorg/modules. 

 **Standard KD:** <userA>'s Serval Performance system has failed to boot several times since installing the latest Ubuntu 'edgy' kernel security update. The problem required a hard shutdown and reboot. The first time, <userA> tried booting into recovery mode, which seemed to work around the issue. The second time, even recovery mode did not work; three times <userA> attempted to boot without success. <userA> noticed the message "BUG: soft lockup on CPU#0" and flipped the wireless card switch on the chassis to the 'on' position. However, he is unsure if the switch was the cause/cure or not. <userB> and <userC> suggested that the problem may have been caused by a CPU bug where one of the cores was always at 100% frequency even when not in use according to the system monitor. The issue seems to be related to coming out of hibernation. If <userA> boots his system fresh, he does not observe the behavior, but if he hibernates and resumes, he notices it. 

 **SPF KD:** <userA> installed the latest Ubuntu 'edgy' kernel security update and the Serval Performance system failed to boot several times. The problem required a hard shutdown and reboot. The user tried booting into recovery mode, which seemed to work around the issue, but the second time it did not work, and the user noticed the message "BUG: soft lockup on CPU#0". The user has just booted the system successfully again after flipping the wireless card switch on the chassis to the 'on' position. However, the user does not know if the switch was the cause/cure or not. The issue seems to be related to coming out of hibernation, and if the user boots the system fresh, they do not observe the behavior, but if they hibernate and resume, then they notice it. <userA> has reinstalled the nvidia-glx package just in case. 

 **Reference:** The user's serval performance system has failed to boot several times. The problem required a hard shutdown and reboot. When the problem occurred first time, the user tried booting into recovery mode, which seemed to work around an issue. When the problem occurred second time, even recovery mode did not work. The user tried booting the system after flipping the wireless card switch on the chassis to the 'on' position. The user noticed the message BUG: soft lockup on CPU#0. The user also noticed that one of the cores was always at 100% frequency, even when not in use according to the system monitor due to the CPU bug. The other user found the report which had mentioned problem due to the same bug and was trying to recreate the problem on a serval. The cause of the problem might be mentioned in this webpage. http://ubuntuforums.org/showthread.php?t=318206 The user had the driver version information. The user felt that the wireless card/driver was not related to the CPU frequency issue but it was related to the system coming out of hibernation. The user found that the system was booted the problem was not observed but when the system was resumed from hibernation, the problem was observed. |

Table 4: Summaries generated by ChatGPT (two-shot, USE-cosine), Standard KD, and SPF KD on an example Ubuntu forum conversation (usernames are anonymized).