# OpenReview forum: "Select, Prompt, Filter: Distilling Large Language Models for Summarizing Conversations"
_EMNLP/2023/Conference — EMNLP 2023 Main_

### Official Review · Reviewer_AQP7 · 2023-08-04

**Soundness:** 4

**Excitement:**

4: Strong: This paper deepens the understanding of some phenomenon or lowers the barriers to an existing research direction.

**Paper Topic And Main Contributions:**

The paper presents an approach for generating weakly supervised annotated datasets for data summarization using knowledge distillation of LLMs (ChatGPT). It addresses the problem of scarcity of annotated training data for finetuning summarization models for a specific domain. The main contributions of the paper include an approach for data efficiency as well as an approach for generating new language training resources.
The proposed approach consists of three steps - select, prompt, and filter, which generate relevant training examples from a big unannotated corpus. The selection phase identifies the most similar instances from the unannotated corpus to the validation examples using cosine similarity of sentence embeddings (Universal Sentence Encoder). Once the examples are selected, the prompt phase generates a few-shot prompt to generate the summary and uses ChatGPT to obtain the summary of the example. In the filtering phase, the synthetic examples that do not satisfy a certain Shannon Score threshold are removed from the dataset.
The paper performs experiments using 3 different forum conversation datasets in the general domain (Ubuntu, NYC, BC3). Using the proposed approach, a distantly supervised dataset is generated and used to train a language model (XSum BART). The authors compare the ROUGE-1/2/L metrics of BART trained using the generated dataset with several other systems - ChatGPT (zero/few-shot), BRIO, and Pegasus. Using the generated dataset, the fine-tuned model shows improved performance over all 3 datasets and metrics.
The authors perform an ablation study for each part of the process - select, prompt, and filter which shows that each part contributes to the improved performance but the improvement varies based on the test set.
The paper presents a practical approach that can be applied to different summarization datasets and domains and can help tackle the problem of insufficient training data. Also, it efficiently uses the resources to generate summaries only for relevant samples from the unannotated dataset. Experiments show using the proposed approach as opposed to a random selection can achieve the same ROUGE-2 score using just half of the number of training examples, which reduces the cost and is more environmentally sustainable.


**Questions For The Authors:**

Question A: When comparing the performance improvements across the three datasets, you notice that the Ubuntu dataset shows the biggest gains. Can we estimate if the data from an unannotated corpus would be a good fit for a given validation set? Is there a significant difference in the average cosine similarities between the closest samples from the unannotated corpus and each of the validation datasets in the selection and prompt phases?
Question B: What embeddings are you using to compute the USE cosine similarity - GloVe?
Question C: Do you have any insights on the quality and factual correctness of the generated summaries?


**Reasons To Accept:**

•	A well-structured and well-written paper.
●	The paper presents an approach for generating weakly supervised datasets for summarization which can be used for more efficient model training and can make up for the lack of training data
●	The authors show another way of leveraging LLMs to distill knowledge and generate annotated datasets
●	The approach can be applied in different domains
●	The approach shows improved results over direct LLM usage and competitive models


**Reasons To Reject:**

●	The paper uses XSum BART for its experiments but it doesn’t specify why this model was selected. Also, there is information about the training parameters but not how they were selected.

**Reproducibility:**

3: Could reproduce the results with some difficulty. The settings of parameters are underspecified or subjectively determined; the training/evaluation data are not widely available.

**Reviewer Confidence:**

3: Pretty sure, but there's a chance I missed something. Although I have a good feel for this area in general, I did not carefully check the paper's details, e.g., the math, experimental design, or novelty.

**Typos Grammar Style And Presentation Improvements:**

The paper is well-written and easy to follow.
Minor suggestions:
In Table 1 it would be helpful if you distinguish between the dataset and model used in the experiment as well as create separate subcolumns for ROUGE-1/2/L.
If space allows, showing an example of a text and summary would be helpful.
It would be very valuable if the authors could publish the code and the generated dataset.

---

> ### Author Rebuttal · Authors · 2023-08-29
>
> Dear Reviewer AQP7,
>
> We thank you for your valuable feedback on our paper. Please find our responses below.
>
> > **_“The paper uses XSum BART for its experiments but it doesn’t specify why this model was selected.”_**
>
> We had tried several pre-trained models (including PEGASUS, T5) in many of our initial explorations with fine-tuning and knowledge distillation for summarization. We found that fine-tuned BART generally performs best and produces good output summaries. A very recent preprint also reports a similar finding (Demeter et al, 2023, https://arxiv.org/pdf/2306.10555.pdf) that shows fine-tuned BART produces better outputs compared to PEGASUS and T5, both in terms of human and automatic evaluation. We will add details regarding our choice and this citation in the revised paper.
>
> > **_“Also, there is information about the training parameters but not how they were selected.”_**
>
> The training parameters were either experimentally set based on validation performance or due to experimental constraints. For instance, we observed overfitting of our models after 5 epochs of finetuning and 100K training examples were chosen due to our ChatGPT annotation budget. We will add more details in the revised paper.
>
> **Answers to questions:**
>
> > **_Question A: “Can we estimate if the data from an unannotated corpus would be a good fit for a given validation set?”_**
>
> Yes, average cosine of each training example across the validation examples can indicate if the unannotated dataset is good for it. For example, we found that Ubuntu, NYC, and BC3 dev sets had an average cosine of 0.7, 0.4, and 0.5, respectively across the selected training examples. Our improvements on these test sets with our selection method also follows the same order (Ubuntu > BC3 > NYC).
>
> > **_Question B: “What embeddings are you using to compute the USE cosine similarity - GloVe?”_**
>
> We use Universal Sentence Embeddings (Cer et al. 2018, https://tfhub.dev/google/universal-sentence-encoder/4) as the sentence representations. We give details about the same in Line 121-126 in the paper (Section 2, ‘USE-Cosine’)
>
> > **_Question C: "Do you have any insights on the quality and factual correctness of the generated summaries?”_**
>
> We inspected some of the conversations and the generated summaries of our system and ChatGPT. Overall, the systems are mostly on par in terms of factual correctness. However, in one example, we noticed that ChatGPT hallucinated information that was not entailed from the input conversation (The conversation is about a person who is planning to visit Buffalo for a football game. However, ChatGPT hallucinated that the conversation was about going to  a game by “Sabres” which is a hockey team from Buffalo). Compared to our system that is tuned for the summarization task, ChatGPT has more generative capabilities given the data it has seen during training. This makes it capable of answering outside-of-context potentially leading to factual inaccuracies. We also noticed that ChatGPT tends to produce shorter summaries than our system leading to the omission of important facts from the summary for some of the conversations.

---

### Official Review · Reviewer_jSRx · 2023-08-06

**Soundness:** 4

**Excitement:**

2: Mediocre: This paper makes marginal contributions (vs non-contemporaneous work), so I would rather not see it in the conference.

**Missing References:**

1. Xu, Can, Qingfeng Sun, Kai Zheng, Xiubo Geng, Pu Zhao, Jiazhan Feng, Chongyang Tao, and Daxin Jiang. "Wizardlm: Empowering large language models to follow complex instructions." arXiv preprint arXiv:2304.12244 (2023).

**Paper Topic And Main Contributions:**

The paper presents an approach for distilling conversation summarization capabilities using the outputs from a large language model (i.e., ChatGPT). The authors propose a three step approach based on: 1/ example (conversation) selection from a set of unlabeled ones, 2/ prompt retrieval (matching the closest prompt to the conversation), and 3/ automatic summary generation and filtering out the inconsistent outputs. The approach is evaluated on three different summarization datasets (Ubuntu, NYC, BC3).

**Questions For The Authors:**

1. How does the diversity of prompts affects the quality of the distilled models?

**Reasons To Accept:**

- The proposed approach improves the results over the teacher model ChatGPT using a substantially smaller model (i.e., BART).
- More automatically produced data leads to higher results, and the model outperforms a standard finetuning baseline on data from an external task.

**Reasons To Reject:**

- Producing training examples from larger models and using them for training smaller ones is known to be effective for many NLP tasks, which limits the novelty of this work.


**Reproducibility:**

3: Could reproduce the results with some difficulty. The settings of parameters are underspecified or subjectively determined; the training/evaluation data are not widely available.

**Reviewer Confidence:**

3: Pretty sure, but there's a chance I missed something. Although I have a good feel for this area in general, I did not carefully check the paper's details, e.g., the math, experimental design, or novelty.

**Typos Grammar Style And Presentation Improvements:**

- Dt is never introduced formally

---

> ### Author Rebuttal · Authors · 2023-08-29
>
> Dear Reviewer jSRx,
>
> We thank you for your valuable feedback on our paper. Please find our responses below.
> > **_“Producing training examples from larger models and using them for training smaller ones is known to be effective for many NLP tasks, which limits the novelty of this work.”_**
>
> Our paper presents a novel pipeline to refine and improve the standard distillation process (which is our baseline). We do not claim that the novelty of our approach lies in directly applying knowledge distillation. Specifically, our paper focuses on the sequential application of three steps in the LLM distillation process that improves summarization performance (see Section 4 of the results) and minimizes the costs of generating the data (see Figure 3) over the standard distillation process.
>
> > **_“How does the diversity of prompts affect the quality of the distilled models?”_**
>
> We explored using 1-shot and 2-shot prompts and found that the performance of ChatGPT increases with increasing the number of priming examples. This results in an improvement in performance of the 2-shot over the 1-shot scenario by +1/+1.5/+1.5 ROUGE-1/ROUGE-2/ROUGE-L scores. We conducted the distillation experiments with 2-shot prompts for ChatGPT assuming a better teacher model results in better training data for the distilled models.
>
> > **_“$D_t$ is never introduced formally”_**
>
> $D_t$ has been introduced in line 151 as a notation for the unannotated training dataset. We will state that it is a set of forum threads.

---

### Official Review · Reviewer_8qtH · 2023-08-12

**Soundness:** 3

**Excitement:**

3: Ambivalent: It has merits (e.g., it reports state-of-the-art results, the idea is nice), but there are key weaknesses (e.g., it describes incremental work), and it can significantly benefit from another round of revision. However, I won't object to accepting it if my co-reviewers champion it.

**Paper Topic And Main Contributions:**


This paper presents a novel approach, "select-prompt-filter knowledge distillation" (SPF KD), aimed at efficiently using large language models (LLMs) for summarizing conversations in low-resource domains. The challenge addressed is the costly fine-tuning of LLMs on specialized tasks. SPF KD involves selecting conversation data similar to a small annotated set, generating prompts for LLMs, and filtering low-quality data. The approach shows significant performance gains in summarization tasks, making it valuable for tasks in resource-scarce domains. Overall, the paper introduces a strategic solution for optimizing LLMs' utility in summarization while minimizing costs.

**Reasons To Accept:**

1. SPF KD offers a novel way to optimize LLMs for low-resource tasks, showing strong summarization improvements.
2. Addresses expensive LLM fine-tuning, providing a cost-effective solution for NLP tasks.

**Reasons To Reject:**

Limited to forum conversations, SPF KD's applicability to diverse text types might be uncertain.


**Reproducibility:**

3: Could reproduce the results with some difficulty. The settings of parameters are underspecified or subjectively determined; the training/evaluation data are not widely available.

**Reviewer Confidence:**

2: Willing to defend my evaluation, but it is fairly likely that I missed some details, didn't understand some central points, or can't be sure about the novelty of the work.

---

> ### Author Rebuttal · Authors · 2023-08-29
>
> Dear Reviewer 8qtH,
>
> We thank you for your valuable feedback on our paper. Please find our responses below.
>
> > **_“Limited to forum conversations, SPF KD's applicability to diverse text types might be uncertain.”_**
>
> Automatic abstractive summarization carries significant value for the conversational domain given the massive usage of online communication channels (forums, chats, emails etc.). We lack large-scale annotated conversational corpora for training summarization models. For this reason (and due to ChatGPT usage costs), we focused our proposed approach and experiments on conversation summarization (mentioned in the Limitations section).  However, it is important to note that we have evaluated our approach on two text types (forums and emails) and two domains (IT and travel).

---

### Meta-Review · Area_Chair_a2GG · 2023-09-19

**Recommendation:** 5

**Metareview:**

The paper brings merits regarding "select-prompt-filter knowledge distillation" and enhance the summarization for low-resource domains, which is valuable. The paper is expected to add more tech details as mentioned in the rebuttal.

---

### Meta-Review · Senior_Area_Chairs · 2023-09-30

**Recommendation:** 4

**Metareview:**

The paper presents an approach for generating weakly supervised annotated datasets for conversation summarization using the outputs from a large language model ; showing strong summarization improvements for low-resource settings. The reviewers agree the paper is sound but have concerns about the incremental nature of the approach and the limited applicability.

---

### Decision · Program_Chairs · 2023-10-07

**Decision:**

Accept-Main

**Comment:**

The paper brings merits regarding "select-prompt-filter knowledge distillation" and enhance the summarization for low-resource domains, which is valuable. The paper is expected to add more tech details as mentioned in the rebuttal.|The paper presents an approach for generating weakly supervised annotated datasets for conversation summarization using the outputs from a large language model ; showing strong summarization improvements for low-resource settings. The reviewers agree the paper is sound but have concerns about the incremental nature of the approach and the limited applicability.